# Facial Feature Extraction Using a Symmetric Inline Matrix-LBP Variant for Emotion Recognition

**DOI:** 10.3390/s22228635

**Published:** 2022-11-09

**Authors:** Eaby Kollonoor Babu, Kamlesh Mistry, Muhammad Naveed Anwar, Li Zhang

**Affiliations:** 1Faculty of Engineering and Environment, Department of Computer and Information Sciences, Northumbria University, Newcastle upon Tyne NE1 8ST, UK; 2Department of Computer Science, Royal Holloway, University of London, Surrey TW20 0EX, UK

**Keywords:** local binary patterns, adaptive image transformation, coded visual descriptors, image encoding, facial expression recognition

## Abstract

With a large number of Local Binary Patterns (LBP) variants being currently used today, the significant and importance of visual descriptors in computer vision applications are prominent. This paper presents a novel visual descriptor, i.e., SIM-LBP. It employs a new matrix technique called the Symmetric Inline Matrix generator method, which acts as a new variant of LBP. The key feature that separates our variant from existing counterparts is that our variant is very efficient in extracting facial expression features like eyes, eye brows, nose and mouth in a wide range of lighting conditions. For testing our model, we applied SIM-LBP on the JAFFE dataset to convert all the images to its corresponding SIM-LBP transformed variant. These transformed images are then used to train a Convolution Neural Network (CNN) based deep learning model for facial expressions recognition (FER). Several performance evaluation metrics, i.e., recognition accuracy rate, precision, recall, and F1-score, were used to test mode efficiency in comparison with those using the traditional LBP descriptor and other LBP variants. Our model outperformed in all four matrices with the proposed SIM-LBP transformation on the input images against those of baseline methods. In comparison analysis with the other state-of-the-art methods, it shows the usefulness of the proposed SIM-LBP model. Our proposed SIM-LBP variant transformation can also be applied on facial images to identify a person’s mental states and predict mood variations.

## 1. Introduction

Classifying facial features from images is a challenging computer vision task. If we improve the reliability and efficacy of this operation, the application scope will be widened in particular in tackling real-world challenges [1]. Initial steps in recognizing facial expression (FE) are to extract the facial features from an image. Local descriptors play a vital role in discriminative feature extraction [2]. This importance has gathered research attention from scientific community in developing new methods with respect to local descriptors. These local descriptors are the fundamental building blocks of the feature extraction process. One of the popular descriptors is the Local Binary Patterns (LBP) [3,4]. It was one of the earliest feature descriptors. After the introduction of LBP, there were many variants of LBP proposed by researchers, which have shaped the field of feature extraction and pattern recognition. However, there were not that many research studies conducted on feature descriptors that are dedicated for facial features, specific for facial expression recognition (FER).

While we look at emotion recognition as a computer vision task, the first question we had was what visual information is critical for an emotion recognition system. This question pointed us to facial action units [5] and its classification with respect to emotions. Therefore, the main motivation for our research was to accurately map facial action units with emotion categories. Taking this as a challenge, we realized that, if we have a dedicated visual descriptor that can focus on the facial action units as a primary target, it will create a system that will be more reliable and faster compared to other traditional emotion recognition models; it is also a light weight model compared to other similar models. We had multiple candidates in place as our visual descriptor, but LBP’s [3] had a unique property of thresholding the neighborhood of each pixel and consider the result as a binary number which makes the computation faster on identifying predefined patterns (in our case, its facial action units); this was one of the main reasons why we decided to make a variant of LBP that will act as the dedicated visual description for emotion recognition.

If we look at the application areas of emotion recognition, one of the main and emerging fields is human–computer interaction. In recent HCI projects [6], there are systems that are created which make it possible for a two-way communication between a machine and a human. In this kind of a setup, it is very vital that the machine understands the appropriate emotion of the human subject it is interacting with to make the overall system more dynamic. Thus, our proposed model plays a key role in such a system, which makes emotion recognition more reliable and faster.

Therefore, we propose a new variant of LBP called the Symmetric Inline Multiplier LBP (SIM-LBP), which is a dedicated feature descriptor for facial expression features. The three main contributions of our proposed feature descriptor are as follows: Firstly, it reduces computation in each of the feature vectors that is being considered owing to the fact that other feature descriptors use 3-pair values, whereas we use a 2-pair value system. In addition, the proposed model can compute more than one feature at the same time parallelly to reduce the total computation time. Finally, it generates a normalized histogram of every cell and provides a look-up window for the required facial feature to be extracted from the image, which makes it a unique feature descriptor for facial features. We also show that our model is capable of detecting features from images suffering from low lighting problems and blurry. Our model showed better results compared to other models when performing similar FER tasks. Extensive experiments are conducted using state-of-the-art models to evaluate model effectiveness.

We have applied the proposed SIM-LBP to transform a well-known Japanese female facial expression dataset (JAFFE) [7]. This dataset has 10 Japanese female subjects with different facial expressions annotated and consists of 213 total images [8]. After the transformation was done on all the images in the above stated dataset, we trained a Convolutional Neural Network (CNN) [9,10] model to predict the facial expression of the subject. The input images to the CNN model were first transformed using our proposed SIM-LBP feature descriptor. We also feed the SIM-LBP transformed image dataset in the training phase. The results from our experiments are showcased in the evaluation section. We also conducted a relative study to showcase the importance of using our SIM-LBP image transformation in FER. 

Our model achieved reliable and efficient performance in comparison with images that do not use the proposed SIM-LBP transformation. We have evaluated our model’s expression recognition accuracy rate, precision, recall, and F1-score metrics with other similar models. In all four metrics, our model achieves better overall performance. The proposed feature descriptor also performed better in comparison with other five state-of-the-art feature descriptions, i.e., LBP, 6 × 6 Multiscale block LBP (MB-LBP) [11], Histogram of Oriented Gradients (HOG) [12,13], Median Binary Patterns (MBP) [14] and Orthogonal difference-LBP (OD-LBP) [15].

This paper is structured as follows: In Section 1, an explanation on the need for facial emotion recognition and its importance is explained. Then, an explanation on the proposed model and its background is provided. Explanation with respect to the facial expression datasets is also provided. The literature review is provided in Section 2. Section 3 describes the proposed SIM-LBP method, and Section 4 reports the evaluation conducted and detailed results in comparison with several existing studies. Section 5 concludes this study and identifies future directions.

## 2. Related Work

LBP is oriented based on the concept of finding the difference between the neighboring and the central pixels, as well as generating a weight regarding this difference. This was the concept that was used in the traditional LBP. This led to many problems while finding the patterns on images with irregular shapes and structures. In addition, the traditional LBP operators are comparatively slow when tackling large-scale images as they produce long histograms. Moreover, the binary data generated by conventional LBP operation are more vulnerable to noise. 

Due to these problems, there are many LBP variants proposed to address all these issues and make LBP more resilient and efficient. In [16], a proposal was made that focuses on extraction and reduction in multiple features. They have carried out a discrete wavelet transformation on RGB channels initially, so approximation and correct coefficients are applied on a channel invariant LBP. This is very useful when we want the result to be rotationally invariant. For every local neighbor 3 × 3 window, a rotation invariant function is generated by estimating the descriptors relative value to a reference value. Their work provides full structural information from the LBP; additionally, it also has the information based on the magnitude of the visual descriptor. Their proposed LBP descriptor is very ideal for texture image classification as texts in an image can be at any given angle.

Most of the existing LBP’s are used for feature extraction in general, but if we look at the FER as a challenge, we have some LBP’s [17] that are intended to extract eyes, eye brows, nose and mouth, bur our model has a better overall detection accurate with less computational resources. Our main objective was to enhance the emotion detection accuracy. Thus, by using our proposed system on an Emotion detection system, the overall detection speed will improve, as the system focuses on the key facial landmark regions of the subject for emotion detection. 

In [18], modified HOG and LBP are used to detect facial emotion of a subject. They used LBP to extract the required features from eyes, nose and mouth regions which were already extracted from the images using HOG. The intention here was to filter out unwanted regions of the selected features using LBP in combination with HOG. 

A blind forensics detection approach that uses a uniform local binary pattern (ULBP) is proposed by [19]. ULBP histogram features were extracted and sent to two different sets of classifiers, which checked whether the image was resized or reshaped. A nonparametric descriptor was used in ULBP to get the binary vector code for both uniform and non-uniform patterns that were assigned to a single bin. 

In [20], a hybrid LBP was proposed, which uses a wavelet-based approach for detecting epilepsy by classifying EEG signal. EEG signal, which is analog in nature, is used to do an LBP calculation. After processing LBP calculation on the entire set data, a sampling is done to reduce the resolution of the initial binary codes; after that, the feature extraction from the signal was complete.

Another method was proposed by [21] that uses ECG signals for emotion detection, and the key technique used in their proposed approach is a combination of automated CNNLSTM with ResNet-152 algorithm. This method provided a higher accuracy score due to the hybrid deep learning approach that is incorporated.

A circular regional mean completed local binary pattern (CRMCLBP) was proposed by [22]. It used a circular mean operator region to alter the properties of a conventional completed local binary pattern (CLBP). It has also two encoded schemes to enhance the feature extraction properties. Their LBP variant performed very well with rotation invariance images, with significant feature representation capability and higher resistance to image noise. 

Three Orthogonal Planes based LBP called Local Binary Pattern from Three Orthogonal Planes (LBP-TOP) [23] showed superior performance for micro expression detection. However, it only extracts the dynamic features in horizontal and vertical directions from an image; thus, to overcome these limitations, Ref. [24] proposed a new variant of LBP called the Five Intersecting Planes LBP (LBP-FIP). This LBP is very efficient in analyzing the facial muscles movements and variations, making it an ideal candidate for micro expression detection from video inputs. Specifically, it used a combination of Eight Vertices LBP (EVLBP) and LBP-TOP that was extracted from three planes. By using this approach, more dynamic features are extracted from the images directly. As it was using all eight-pixel values of a 3 × 3 window, the overall recognition time was impacted. However, the model performed very well with features in the oblique directions. 

In [17], to enhance the discriminative nature of LBP, a horizontal and vertical neighborhood pixel comparison LBP (hvnLBP) was proposed. It was used in combination with Gabor filter for improved performance. This algorithm has four main steps to implement feature selection, starting with an illumination correction and noise filtering in the first step, followed by face detection for getting region of interest (ROI). In the third phase, it used a Gabor filter technique on the facial image. After these three pre-processing steps, the proposed hvnLBP was applied. The proposed hvnLBP showcased competitive ability in expressing discriminative contrast data in the image. This made the proposed method perform efficiently when extracting corners and edges in an image.

In [25], an enhanced deep clustering network is proposed that is developed in four different stages. In stage one, a feature extractor is used to obtain the most prominent key features from the input; then, a conditional generator is used to create a condition according to the parameters set; this is then passed to a discriminator network that constantly iterates till the model does a real joint distribution from the latent representations for features extracted from the previous stage to further enhance it. Finally, when the condition is met, the Siamese network detects the embedding space, so a better affinity similarity matrix is generated.

A new method is used to reformulate the recently developed fractional programing by reducing the feature redundancy in [26]. The key advantage of this method is that it used bilevel and bilinear programing which helps in avoiding the use of fractional function that is very computation regressive. This method is a neurodynamic approach that yields a higher classification performance on six benchmark datasets.

If we look at some of the recent surveys in facial expression recognition, it is very evident that emotion recognition is still a challenging computer vision task that has many different challenges. While most of the traditional emotion recognition system can only detect the expression from an image that has front facial angles, in recent surveys [27,28,29], it is very clear that many new models are proposed that can even detect emotions from images with different face angles. In addition, it is very clear from the above stated surveys that publication in emotion recognition is rapidly increasing if we look at the last 10-year pattern.

## 3. The Proposed LBP Variant

Most of the local descriptors used today are based on the relationship between the neighborhoods pixels and the central pixel by utilizing the 3 × 3 pixel window. Moving one step forward from the previous methodologies used, we introduce the novel descriptor for facial expression and illumination variations called Symmetric Inline Matrix LBP (SIM-LBP). The detailed description of the proposed descriptor is given below (Table 1).

In SIM-LBP, two gray level differences are computed for each of three center vertical and horizontal positions of a 3 × 3 pixel window, by subtracting the two closest left and right neighborhood pixel values from the center one. The diagram explaining this process in shown in Figure 1. Let us consider Figure 1a below that denotes the pixel position notations. SL1 is the top leftmost pixel, SL2 is the one below SL1 and SL3 is the left bottom pixel. Similarly, SC1, SC2 and SC3 are the corresponding pixel positions for the three center pixels. SR1, SR2 and SR3 are the three vertical right most pixels. To explain the working of our proposed algorithm, we have used a sample 3 × 3 pixel window as shown in Figure 1b. In the first iteration, a 3 × 1 vertical window is selected from the center of the 3 × 3 window as shown in Figure 1c. Then, we subtract the center value from the left and right pixel values to obtain a pair of values for each of the center 3 × 1 pixels. After this task is completed on all three center vertical locations, we obtain a resultant set of values as shown in Figure 1e. Similarly, we do the same process for the 1 × 3 center horizontal pixels (SL2, SC2, SR2). Value in SL2 is subtracted from SL1 first and then value in SL2 is subtracted from SL3 to obtain the first pair horizontal values. The same task in performed on SC2 and SR2. The resultant values are shown in Figure 1f. This 2-value pair is called a vertical and horizontal comparison pair. Once these two sets of values are obtained, we used the formula stated in Equation (1) below to generate a feature coefficient value for each of the six horizontal and vertical pixels (three vertical and three horizontal positions). The calculations for obtaining feature coefficients values for SC1, SC2, SC3, SL2, SC2 and SR2 are shown in Figure 1g. Then, by deploying the formula stated in Equation (2) that finds the difference between vertical value pair and the corresponding feature coefficient value, we obtain three sets of vertical binary pairs as shown in Figure 1h. The same task is performed on the horizontal value pair to obtain the horizontal binary pair as illustrated in Figure 1(i). The complete vertical and horizontal binary pairs are shown in Figure 1j,k, respectively. The next step is to perform a symmetric binary split. This process separates the binary pair to the left and right cells of the 3 × 3 matrix for the vertical binary pair and top and bottom cell for the horizontal binary pair. The vertical binary split is shown in Figure 1l, and the horizontal binary split is shown in Figure 1m. In the final stage, an OR operation is carried out on these two binary split matrices, resulting in a merged binary matrix as shown in Figure 1h. In this OR operation, each of the cells in 3 × 3 vertical matrix is evaluated with the corresponding cell in the 3 × 3 horizontal matrix. Then, to obtain the binary pattern from the resultant matrix, each of the binary elements are taken in a clockwise direction starting from the top left most cell. Figure 1o shows the final binary patter attained. The last step is to obtain the weighted pixel value of the generated binary pattern. This is conducted by converting the 8-bit binary pattern code to its corresponding decimal value by using Equation (3). In the example showcased in Figure 1, the weighted pixel value is 106 aka feature size of the SIM-LBP.
(1)$$fcv^{k}=\frac{\sum_{k=1}^{n}\left(x_{k},y_{k}\right)}{\sigma 2\left(x_{k},y_{k}\right)};\;\mathrm{where}\;n=6\;\mathrm{and}\;\sigma 2\;\mathrm{is}\;\mathrm{the}\;\mathrm{variance}\;\mathrm{calculator}$$
(2)$$\left\{\begin{array}{c} Vbp_{1}=\left(\begin{array}{c} \left(V_{x1}-SC1\right)\geq 0\;\;\;;\;\;1\\ \left(V_{y1}-SC1\right)<0\;\;\;;\;\;0 \end{array}\right)\;\;,\;\;Hbp_{1}=\;\left(\begin{array}{c} \left(H_{x1}-SL2\right)\geq 0\;\;\;;\;\;1\\ \left(H_{y1}-SL2\right)<0\;\;\;;\;\;0 \end{array}\right)\\ Vbp_{2}=\left(\begin{array}{c} \left(V_{x2}-SC2\right)\geq 0\;\;\;;\;\;1\\ \left(V_{y2}-SC2\right)<0\;\;\;;\;\;0 \end{array}\right)\;\;,\;\;Hbp_{2}=\;\left(\begin{array}{c} \left(H_{x2}-SC2\right)\geq 0\;\;\;;\;\;1\\ \left(H_{y2}-SC2\right)<0\;\;\;;\;\;0 \end{array}\right)\\ Vbp_{3}=\left(\begin{array}{c} \left(V_{x3}-SC3\right)\geq 0\;\;\;;\;\;1\\ \left(V_{y3}-SC3\right)<0\;\;\;;\;\;0 \end{array}\right)\;\;,\;\;\;Hbp_{3}=\;\left(\begin{array}{c} \left(H_{x3}-SR2\right)\geq 0\;\;\;;\;\;1\\ \left(H_{y3}-SR2\right)<0\;\;\;;\;\;0 \end{array}\right) \end{array}\right\}$$
(3)$$WPV\left(\;weighted\;pixel\;value\right)=\overset{7}{\underset{i=0}{\sum }}2^{i}\times \;fbp\left(i,\left(i+1\right)\right)$$

Then, the above stated SIM-LBP pixel transformation is carried out on all pixels of the images to generate an output image as shown in Figure 2a (output from SIM-LBP). As seen in the output image (Figure 2a), the SIM-LBP transformed images have more enhanced facial landmark boundaries in contrast with the original LBP output images that are shown on the right-hand side (Figure 2b). In addition, for facial emotion detection, it is very important to understand the variations in the pixel positions of eyebrows, eyes, and mouth from a set reference point. It is with this variation that the model will predict the correct facial emotion. By using our proposed SIM-LBP, the facial landmark and its neighboring pixels become more enhanced with respect to its contrast as seen in Figure 2a. Thus, this makes our model have more facial emotion information rich data compared to other traditional methods. 

## 4. Model Evaluation

In the second phase of our project, we transformed images in the respective testing datasets to its corresponding SIM-LBP transformed images. The two datasets that we used in our experiments are JAFFE [30] and CK+ [31]. After the transformation is completed on all the images in the respective dataset, we train a Multibranch Cross-Connection CNN proposed by [32] to predict the facial emotion. This model was an ideal candidate for testing our SIM-LBP as it uses residual connection, Network in Network, and tree structure approaches together to enhance the prediction and make the model dynamic in nature. All the results obtained are provided in this section of the paper. The overall model showcased improved recognition rates in comparison with other state-of-the-art models. This was mainly achieved by enhancing facial expression landmarks by using the proposed SIM-LBP.

The details of the adopted dataset are provided in Table 2 below. The main two datasets that were used for testing were JAFFE and CK+ as stated earlier. In JAFFE, we had 10 classes with each class having 21 subjects, and the image resolution was 256 × 256 pixels. In addition, for CK+ dataset, we had eight classes in total, with each class consisting of 25 images. Image resolution was 640 × 490, and these images were taken from video sequences.

To evaluate the classification performance of our model, we have used two confusion matrices that are shown in Figure 3a,b below, i.e., one of the confusion matrixes shows the classification results using the JAFFE dataset while the other using the CK+ dataset. It is clear from these figures that, while using the JAFFE dataset, subjects showing the Happy emotion had achieved the highest score of 93%, followed by Sad (91%), Surprise (90%), Neutral (88%), Disgust (81%), Fear (79%) and Angry (78%). If we take a look at the results from CK+ dataset, the scores have improved slightly for most of the emotions. The accuracy rate of 97% for the Neutral emotion was at the top of the table. The scores for emotions such as Fear, Happy and Sad fall under the 88–89% bucket. The accuracy rate for Surprise was 79% followed by Disgust (77%) and Angry (76%). While comparing the overall emotion classification accuracy while using JAFFE dataset and CK+ dataset, JAFFE dataset gave an overall emotion classification score of 85.71% and, for CK+ dataset, it was 84.86%. Therefore, when we compare both of the results, there is not much difference between using JAFFE and CK+ datasets. This shows the homogenous nature of the proposed algorithm.

We also performed classification performance analysis by measuring precision, recall, F1-score and accuracy. The results from these tests are shown in Figure 4a–d below. While analyzing the precision score in Figure 4a, when using the JAFFE dataset, the results are: Anger (97%), Disgust (92%), Fear (91%), Happy (97%), Sad (95%), Surprise (91%) and Neutral (97%). The precision score was slightly in the lower bracket while using CK+ dataset, starting with Anger (95%), Disgust (88%), Fear (91%), Happy (91%), Sad (91%), Surprise (89%) and Neutral (90%). Equation (4) defines the formula with respect to precision calculation:(4)$$pr=\frac{True\;Positive}{True\;Positive+False\;Positive}\;;\;where\;pr\;is\;the\;precision\;score$$

Recall scores can be seen in Figure 4b below. While using JAFFE, we obtained 91% for Anger, 97% for Disgust, 95% for Fear, 97% for Happy, 95% for Sad, 91% for Surprise and 97% for Neutral. Similarly, while using CK+ dataset, we achieved 91% for Anger, Disgust and Happy emotions, 89% for Fear, sad and Surprise emotions, while 90% for Neutral. The formula used to calculate the Recall rate is provided in Equation (5):(5)$$rc=\frac{True\;Positive}{True\;Positive+False\;Negative}\;;\;where\;rc\;is\;the\;recall\;rate$$

Figure 4c shows the F1 score achieved while using JAFFE and CK+ datasets of the proposed algorithm. When comparing the F1 scores with respect to both datasets, JAFFE has an average score of 92.57% while CK+ has an average score of 93.86%. While analyzing the individual emotion classes from JAFFE, we achieve 91% for Anger, 89% for Disgust and Fear, 90% for Happy, 97% for Sad and Neutral and 95% for Surprise. In addition, for CK+ dataset, we obtained 91% for Anger, Happy and Surprise, 97% for Disgust and Neutral, 95% for Fear and Sad emotions. To calculate the F1 Score, we use Equation (6) stated below:(6)$$f=\frac{2\times pr\times rc}{2\times \left(True\;Positive+False\;Positive+False\;Negative\right)}\;;\;f\;\mathrm{is}\;F1\;\mathrm{Score}$$

Accuracy was measured for all emotion classes, to understand the dynamic nature of the proposed algorithm. The accuracy test was performed on both the JAFFE and CK+ datasets. The results are shown in Figure 4d below. Looking at the results attained while using JAFFE dataset, Anger, Fear and Happy expressions had the lowest score of 89%. Disgust and Sad expressions were classified with an accuracy rate of 91%, and 95% for Neutral. The highest score was for the Surprise emotion class with an accuracy score of 97%. If we look at the results from CK+ dataset, we obtained 91% for Fear, 95% for Disgust and Sad and Neutral with the highest score of 97% for Anger, Happy and Surprise emotion classes. Equation (7) below shows the formula used for accuracy calculation:(7)$$A=\frac{True\;Positive+True\;Negative}{True\;Positive+False\;Positive+False\;Negative+True\;Negative}$$

We also conducted a very comprehensive evaluation with respect to the Accuracy score of other similar facial expression algorithms. The results from this test are tabulated in Table 3 and Table 4. Table 3 shows the accuracy score while using JAFFE dataset, and Table 4 shows the result while using CK+ dataset. For the evaluation of the JAFFE dataset, Ref. [33] used GoogleNet where the emotion detection accuracy rate was 65.79%, followed by [34], which employed VGG-Face and achieved the accuracy rate of 69.76%. In [35], a DenseNet model was used, and the accuracy was 71.02%. An accuracy rate of 71.52% was obtained by MBCC-CNN.WMCNN-LSTM [36] achieved the best result among the baseline methods with an accuracy score of 95.12%. However, our model outperformed it with an accuracy score of 96.82%.

While considering the CK+ dataset, the lowest score of 70.02% was obtained by a model proposed by [35]. Then, 72.69% was the score that was achieved from the model by [33]. VGG-Face proposed by [34] scored 88.12%. In addition, the model purposed by [36] achieved a score of 97.50%, while using CK+ dataset MBCC-CNN [32] obtained the best result of 98.48% in comparison with those of other baseline methods. With an accuracy score of 98.72%, the proposed SIM-LBP model topped the table on accuracy score.

Table 5 below shows the different hypermeters used on the models used for evaluation, while comparing the accuracy scores of the models CK+ dataset shows better performance on most of the models.

While observing the overall evaluation statistics, we can see that our proposed SIM-LBP model outperformed many state-of-the-art models with respect to multiple parameters. This shows the models dynamic nature. While looking at the output generated from our model, we can observe that the model is capable of highlighting the facial landmark region, which is vital in obtaining the emotion readout accurate. This was one of the main reasons that our model was performing well when using the Multibranch Cross-Connection CNN.

We have also provided the ROC graphs for six emotion classes in Figure 5 below. Happy, fear, surprise and disgust have shown better classification performance, whereas sad and angry have slight variation in some of the images that are being classified; this is mainly due to the fact that sad and anger have many common facial action units. This can be addressed with a slight variation in the optimization parameters.

The output from the SIM-LBP is shown in Figure 6 below. Figure 6a shows the seven emotion outputs generated using JAFFE dataset. It is very clear from this output image that the facial landmark and the surrounding regions are more distinct with respect to the original image. Figure 6b shows the seven emotion classes obtained from CK+ dataset. Even though the CK+ dataset uses images extracted from video sequences, the outputs generated are showing the same feature of providing a more segregating nature of facial landmark and its surrounding regions. Therefore, we can conclude that our proposed SIM-LBP model performs sufficiently well on both still images and video data.

## 5. Conclusions

In this research, we have proved that, by using our proposed Symmetric Inline Matrix LBP on the CK+ and JAFFE datasets, we have been able to precisely and resourcefully carry out feature extraction from facial images to aid subsequent facial expression recognition. Our model reduced the need for performing system preprocessing steps like contrast enhancing, gradient operators or dimensionality reduction, to obtain more reliable and efficient facial expression recognition results. The proposed model has shown improved recognition accuracy scores as compared with those of many other similar models. Our comparison study shows that our technique generates better test scores compared to the similar models proposed in the last 6 years. With further enhancements to our model, SIM-LBP can be used for generating more realistic facial emotions for characters in games and animated movies.

However, our model has limitations while performing the SIM-LBP operations on very low-resolution images and videos as well as on very high-resolution images and videos. This is due to fact that extracting the appropriate facial landmark boundary from low resolution images is still a challenge. We are planning to further investigate this challenge, so the noise sensitivity of the model can be improved, which will help the model’s ability to capture more discriminative features even from high and low-resolution images.

Currently, more complex and expensive hardware are required to generate realistic facial expressions on these characters. In particular, Real-time Facial Motion Capture framework uses videos recorded from a small, helmet-mounted camera to record the actor’s facial performance in real time. Thus, to overcome this barrier, we can train a generative model with the results obtained from our proposed SIM-LBP model to generate more realistic synthesized facial emotion images for animated characters.

## Figures and Tables

**Figure 1 sensors-22-08635-f001:**
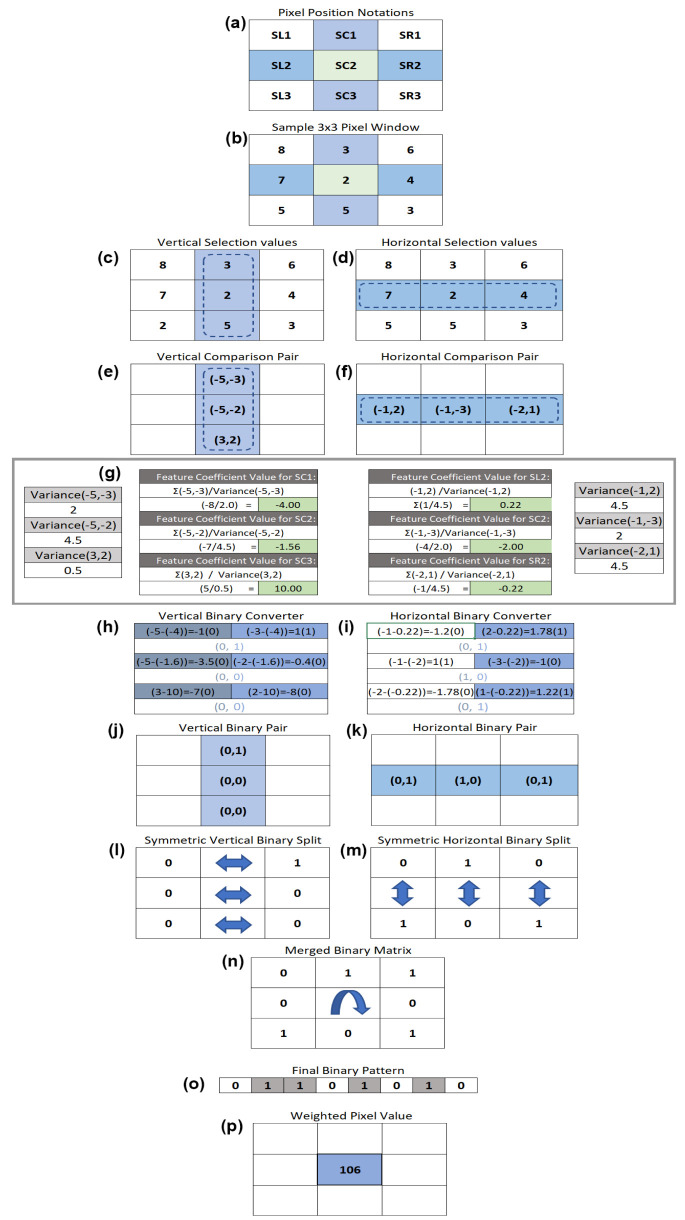
Proposed SIM-LBP approach. (**a**) Pixel position notations of a 3 × 3 pixel window, (**b**) Sample 3 × 3 pixel window, (**c**) Vertical selection pixel values from the 3 × 3 window(highlighted in deep blue), (**d**) Horizontal selection pixel values from the 3 × 3 window(highlighted in light blue), (**e**) Vertical comparison pair, (**f**) Horizontal comparison pair, (**g**) Variance and feature coefficient matrix with calculations, (**h**) Vertical binary conversion, (**i**) Horizontal binary conversion, (**j**) Vertical binary pair, (**k**) Horizontal binary pair, (**l**) Vertical binary split, (**m**) Horizontal binary split, (**n**) Merged binary matrix, (**o**) Flattened binary array, (**p**) Weighted pixel value.

**Figure 2 sensors-22-08635-f002:**
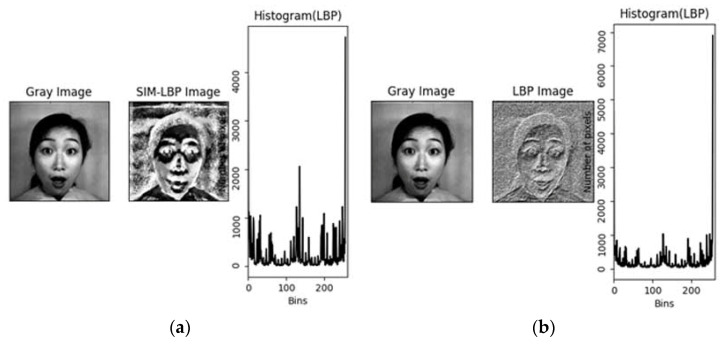
(**a**) SIM-LBP output; (**b**) normal LBP output.

**Figure 3 sensors-22-08635-f003:**
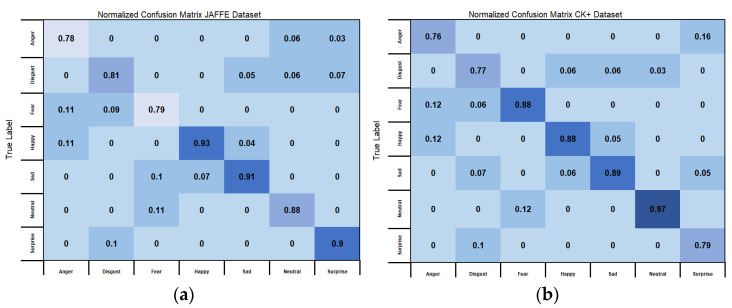
(**a**) Confusion matrixes for the JAFFE dataset; (**b**) confusion matrixes for CK+ Dataset.

**Figure 4 sensors-22-08635-f004:**
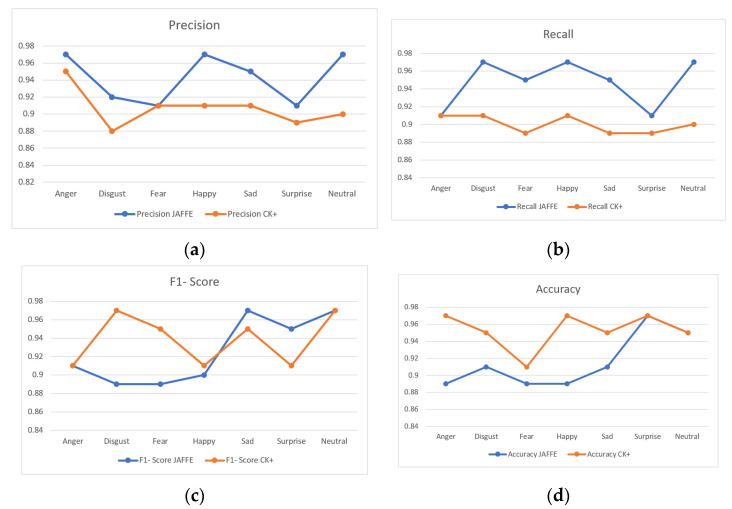
(**a**) Precision score; (**b**) recall score; (**c**) F1-score; (**d**) accuracy.

**Figure 5 sensors-22-08635-f005:**
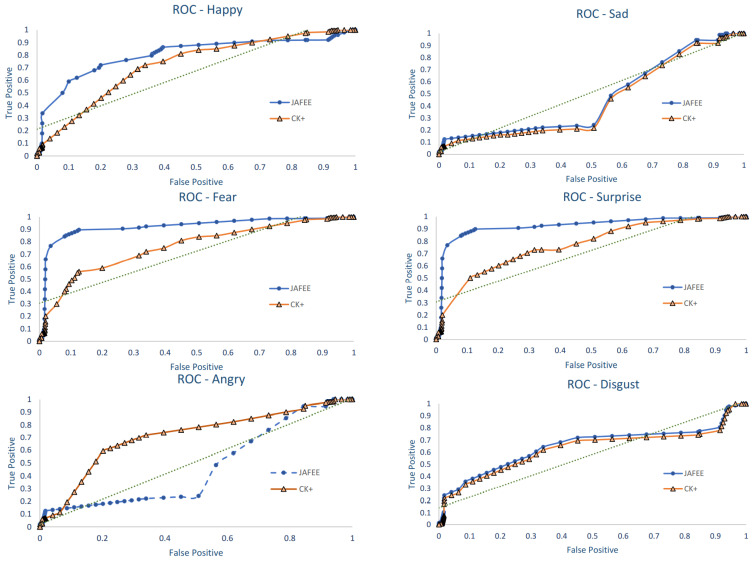
ROC graphs for six basic emotions.

**Figure 6 sensors-22-08635-f006:**
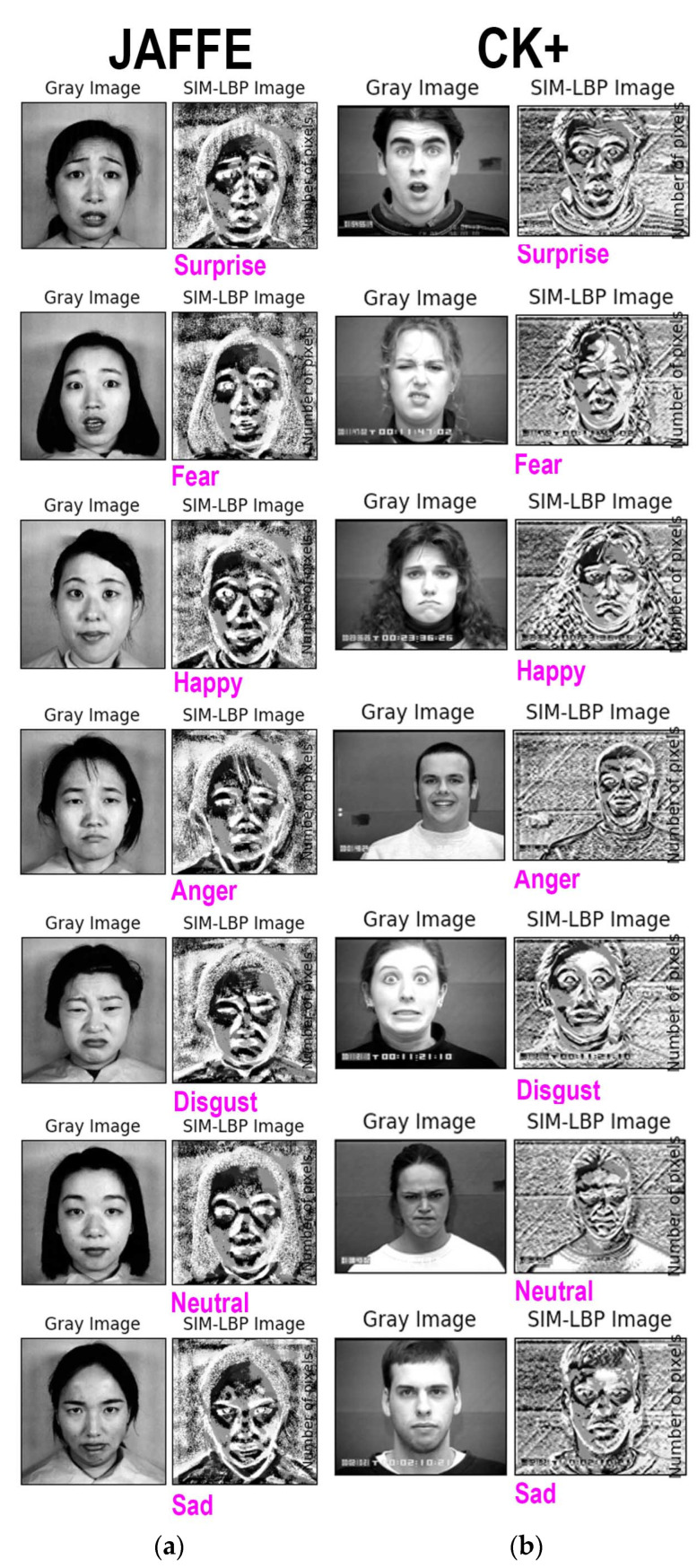
(**a**) Output from JAFFE dataset; (**b**) output from CK+ Dataset.

**Table 1 sensors-22-08635-t001:** SIM-LBP work flow description.

Step 1	Take a 3 × 3 pixel window from the input image (Figure 1a).
Step 2	From the 3 × 3 pixel window, select the center vertical pixels and center horizontal pixels.
Step 3	Create a new pair of values from the vertical and horizontal selections; for this, find the difference between the center value with respect to left and right pixel values (shown in Figure 1e,f).
Step 4	Find the variance for all vertical and horizontal pairs.
Step 5	Find the feature coefficient value using Formula (1) for all pairs in both horizontal and vertical selections (Figure 1g).
Step 6	Then, by applying Formula (2) to the resultant value pairs from step 5, generate vertical and horizontal binary pairs.
Step 7	Merge both horizontal and vertical binary pairs in a clockwise pattern of a 3 × 3 pixel window generate an 8-bit binary code using Formula (3) (Figure 1o).
Step 8	Update the weighted pixel value from step 7 with the old value in the input image.
Step 9	Repeat Step 1 to Step 8 for all pixels in the image.

**Table 2 sensors-22-08635-t002:** Details of the dataset used.

Dataset	No. of Classes	Image/Class	Image Resolution	Channels	Total Images
JAFFE	10	21	256 × 256	Single	210
CK+	8	25	640 × 490	Single	200

**Table 3 sensors-22-08635-t003:** Accuracy score comparison using JAFFE dataset.

Method	Accuracy
GoogLeNet [33]	65.79%
VGG-Face [34]	69.76%
DenseNet [35]	71.02%
MBCC-CNN [32]	71.52%
WMCNN-LSTM [36]	95.12%
**SIM-LBP(Proposed Model)**	**96.82%**

**Table 4 sensors-22-08635-t004:** Accuracy score comparison using CK+ Dataset.

Method	Accuracy
DenseNet [35]	70.02%
GoogLeNet [33]	72.69%
VGG-Face [34]	88.12%
WMCNN-LSTM [36]	97.50%
MBCC-CNN [32]	98.48%
**SIM-LBP(Proposed Model)**	**98.72%**

**Table 5 sensors-22-08635-t005:** Hypermeters used on models.

Models	Hyperparameters Used		Accuracy on JAFFE	Accuracy on CK+
GoogLeNet [25]	layers	3	65.79	70.02
	Neurons	512		
	Learning rate	0.1		
VGG-Face [26]	layers	8	69.76	72.69
	Neurons	256		
	Learning rate	0.01		
DenseNet [27]	layers	5	71.02	88.12
	Neurons	256		
	Learning rate	0.01		
MBCC-CNN [24]	layers	8	71.52	97.5
	Neurons	256		
	Learning rate	0.1		
WMCNN-LSTM [28]	layers	4	95.12	98.48
	Neurons	256		
	Learning rate	0.01		
**SIM-LBP (Proposed Model)**	layers	5	96.82	98.72
	Neurons	256		
	Learning rate	0.01		

## Data Availability

The datasets employed in this study are publicly available at the sites of Japanese Female Facial Expressions (JAFFE) Database of digital images (http://www.kasrl.org/jaffe.htm, accessed on 15 July 2022) and the CK+ Database (MER160@pitt.edu).

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
