# Peer review of "Facial Feature Extraction Using a Symmetric Inline Matrix-LBP Variant for Emotion Recognition"

_sensors, 2022, doi:10.3390/s22228635_

Round 1

Reviewer 1 Report

Facial Feature Extraction Using a Symmetric Inline Matrix-LBP Variant for Emotion Recognition

This paper proposes an enhanced version of LBP as a Symmetric Inline Matrix LBP (SIM-LBP) to describe facial features.
The idea is simple but exciting. However, the paper needs to revise very well.

The introduction is very short, the motivation should be clarified, problem statement and the main challenge should be defined.
Therefore it needs to improve. Several very recent surveys missed and could be useful, "Multiview Facial Expression Recognition, a Survey; Facial Expression Recognition: A Survey;Deep Facial Expression Recognition: A Survey".

The methodology needs to clarify. Where the idea comes from? And why these sorts of computations should provide such output?
I suggest showing the overall structure of your model as a diagram.

Evaluations from page 6 are better separated as a particular section (e.g., Section 4).

Author Response

This paper proposes an enhanced version of LBP as a Symmetric Inline Matrix LBP (SIM-LBP) to describe facial features.
The idea is simple but exciting. However, the paper needs to revise very well.

The introduction is very short, the motivation should be clarified, problem statement and the main challenge should be defined.

Updated as suggested in the paper

Therefore it needs to improve. Several very recent surveys missed and could be useful, "Multiview Facial Expression Recognition, a Survey; Facial Expression Recognition: A Survey;Deep Facial Expression Recognition: A Survey".

Updated as suggested in the paper.

The methodology needs to clarify. Where the idea comes from? And why these sorts of computations should provide such output?
I suggest showing the overall structure of your model as a diagram.

Figure 1 shows the overall process flow in sequence from (a-p) of the figure.

Evaluations from page 6 are better separated as a particular section (e.g., Section 4).

Done on the paper as suggested

Reviewer 2 Report

Paper presents a good study on the Facial Feature Extraction Using a Symmetric Inline Matrix -LBP 2 Variant for Emotion Recognition. I believe there is work to be done before it is ready for publication. Most importantly the authors should look into the following aspects:

The proposed method seems to be efficient and good. The logical description is also appreciated but without proper implementation it is hard to rely on proposed method. So authors advised to add the implementation justification.

There are lots of techniques available for feature extraction how Symmetric Inline Matrix -LBP is superimpose on those?

Authors have used SIM-LBP on the JAFFE dataset to convert all the im-20 ages to its corresponding SIM-LBP transformed variant. What if the database is biased as per the algorithm? Have you test this on some another dataset?

# What is the motivation of this paper?

# What is the contribution and novelty of this paper?

#What is the advantage of this paper?

#Which evaluation metrics did you used for comparison?

Author Response

Paper presents a good study on the Facial Feature Extraction Using a Symmetric Inline Matrix -LBP 2 Variant for Emotion Recognition. I believe there is work to be done before it is ready for publication. Most importantly the authors should look into the following aspects:The proposed method seems to be efficient and good. The logical description is also appreciated but without proper implementation it is hard to rely on proposed method. So authors advised to add the implementation justification.

There are lots of techniques available for feature extraction how Symmetric Inline Matrix -LBP is superimpose on those?

Our proposed SIM-LBP is a dedicated feature descriptor for facial expression features. The three main contributions of our proposed feature descriptor are as follows.

  • It reduces computation in each of the feature vectors that is being considered owing to the fact that other feature descriptors use a 3-pair values, whereas we use a 2-pair value system.
  • The proposed model can compute more than one feature at the same time parallelly to reduce the total computation time.
  • Finally, it generates a normalized histogram of every cell and provides a look-up window for the required facial feature to be extracted from the image which makes it a unique feature descriptor for facial features.

Our model is capable of detecting features from images with low lighting conditions and blur. Our model showed better results compared to other models when performing similar FER tasks.

Authors have used SIM-LBP on the JAFFE dataset to convert all the im-20 ages to its corresponding SIM-LBP transformed variant. What if the database is biased as per the algorithm? Have you test this on some another dataset?

            Yes, in the second phase of our experiment (Section 4), we have tested our model with CK+ dataset as well to test the model’s biasing behaviour. And the model showed same performance benchmark even with CK+ dataset, which ruled out the possibility of a biasing nature.

# What is the motivation of this paper?

The main motivation for this paper was that, all most all the existing LBP’s were used for feature extraction in general, but if we look at the FER as a challenge, we do not have any LBP’s that are designed to extract eyes, eye brows, nose and mouth in particular. Our main objective was to get that gap bridged to enhance the emotion detection system. By using our proposed system on a Emotion detection system, the overall detection speed will improve as the system focus on the key facial landmark regions for emotion detection. We have updated this information in the paper as suggested in the Introduction section.  

# What is the contribution and novelty of this paper?

The most prominent novelty/contribution of this paper is that it uses fewer calculation to get the features extracted from a facial image. This is achieved as the feature extraction technique used, focuses on only the key facial landmarks. This improves the detection time and uses lower system resources for computation.

#What is the advantage of this paper?

This paper will help researchers in Emotion detection experiments to enhance their emotion detection accuracy and also reduce the detection time and system resource used for computation.

#Which evaluation metrics did you used for comparison?

The Key metrics used are listed below.

  • Precision Score;
  • Recall Score;
  • F1- Score;
  • Detection Accuracy;
  • Detection time

We have evaluated the model with other state of the art models for emotion detection with and without SIM-LBP transformation.

Reviewer 3 Report

This paper focuses on facial feature extraction by proposing a new SIM-LBP model. Experimental results show superior performance of the proposed model. Followings are my concerns:

1. There are two equation (1). Please double check.

2. The equation on page 8 seems incomplete (???? ????????+). 

3. It is suggested to add Pseudocode/flow chart in order to show the proposed approach more clearly.

4. To show the importance of the research topic, some related works can be reviewed and added as literature review. For example: EEG-based Emotion Recognition Using Hybrid CNN-LSTM Classification; Unsupervised discriminative feature learning via finding a clustering-friendly embedding space; Two-timescale neurodynamic approaches to supervised feature selection based on alternative problem formulations

5. The parameters/hypermeters of the compared approaches should be shown clearly in the paper. 

6. Please capitalize the first letter of the conference names in references (e.g., Ref [3]). 

Author Response

This paper focuses on facial feature extraction by proposing a new SIM-LBP model. Experimental results show superior performance of the proposed model. Followings are my concerns:

  1. There are two equation (1). Please double check.

Equation (1) is just a single equation with variable specification next to it.

  1. The equation on page 8 seems incomplete (???? ????????+). 

While pasting the equation (7) from the writing draft missed half of the equation, Sorry about that.

The full equation used is updated now.

  1. It is suggested to add Pseudocode/flow chart in order to show the proposed approach more clearly.

SIM-LBP Algorithm is added as suggested, also a process flow diagram is provided in figure 1 of the paper.

  1. To show the importance of the research topic, some related works can be reviewed and added as literature review. For example: EEG-based Emotion Recognition Using Hybrid CNN-LSTM Classification; Unsupervised discriminative feature learning via finding a clustering-friendly embedding space; Two-timescale neurodynamic approaches to supervised feature selection based on alternative problem formulations

Added the suggested literature to the paper.

  1. The parameters/hypermeters of the compared approaches should be shown clearly in the paper. 

Added the hypermeters used as advised in table 4 on the paper.

  1. Please capitalize the first letter of the conference names in references (e.g., Ref [3]).

Corrected as advised.

Round 2

Reviewer 1 Report

All of my concerns are addressed and I think the current version is appropriate for publication. Congratulation to the authors.

Author Response

Thanks a lot for your valuable time and for providing us with constructive feedback. It helped us to improve our paper a lot. 

Reviewer 2 Report

The mathematical equation are not written correctly. The author should check them very carefully. The author advised to include the ROC. The author also advised to include the Limitations of proposed techniques. 

Author Response

Comment: The mathematical equation is not written correctly. The author should check them very carefully.

Response: Updated Equation 7 as advised.

Comment: The author advised to include the ROC.

Response: Added ROC graph as advised.

Comment: The author also advised to include the Limitations of proposed techniques.

Response: Updated as advised in the Conclusion section of the paper.